# AN UNDETECTABLE WATERMARK FOR GENERATIVE IMAGE MODELS

**Sam Gunn**[*]
UC Berkeley

**Xuandong Zhao**[*]
UC Berkeley

**Dawn Song**
UC Berkeley

## ABSTRACT

We present the first undetectable watermarking scheme for generative image models. *Undetectability* ensures that no efficient adversary can distinguish between watermarked and un-watermarked images, even after making many adaptive queries. In particular, an undetectable watermark does not degrade image quality under any efficiently computable metric. Our scheme works by selecting the initial latents of a diffusion model using a pseudorandom error-correcting code (Christ and Gunn, 2024), a strategy which guarantees undetectability and robustness. We experimentally demonstrate that our watermarks are quality-preserving and robust using Stable Diffusion 2.1. Our experiments verify that, in contrast to *every prior scheme* we tested, our watermark does not degrade image quality. Our experiments also demonstrate robustness: existing watermark removal attacks fail to remove our watermark from images without significantly degrading the quality of the images. Finally, we find that we can robustly encode 512 bits in our watermark, and up to 2500 bits when the images are not subjected to watermark removal attacks.

## 1 INTRODUCTION

As AI-generated content grows increasingly realistic, so does the threat of AI-generated disinformation. AI-generated images have already appeared online in attempts to influence politics (Ryan-Mosley, 2023). Watermarking has the potential to mitigate this issue: If AI providers embed watermarks in generated content, then that content can be flagged using the watermarking key. Recognizing this, governments have begun putting pressure on companies to implement watermarks (Biden, 2023; California State Legislature, 2024; European Union, 2024). However, despite an abundance of available watermarking schemes in the literature, adoption has remained limited (Seetharaman & Barnum, 2023). There are at least a few potential explanations for this.

First, some clients are willing to pay a premium for un-watermarked content. For instance, a student using generative AI for class might find a watermark problematic. Any company implementing a watermark could therefore put itself at a competitive disadvantage.

Second, existing watermarking schemes noticeably degrade the quality of generated content. Some schemes guarantee that the distribution of a *single* response is unchanged, but introduce correlations across generations.[1] While this might be acceptable for small models, it is questionable whether anyone would be willing to use such a watermark for a model that cost over $100 million just to train (Knight, 2023). In other words, given the vast effort put into optimizing models, any observable change in the model's behavior is probably unacceptable.

**Undetectable watermarks.** *Undetectability*, originally defined in the context of watermarking by Christ et al. (2024), addresses both of these issues. For an undetectable watermark, it is computationally infeasible for anyone who doesn't hold the detection key to distinguish generations with the watermark from generations without — even if one is allowed to make many adaptive queries. Crucially, an undetectable watermark provably preserves quality under any efficiently-computable metric, including quality metrics that are measured across many generations (such as FID (Heusel

---

[*]Equal contribution. Email addresses: `gunn@berkeley.edu`  `xuandongzhao@berkeley.edu`

[1]One scheme with this guarantee is to fix the randomness of the sampling algorithm, so that the model can only generate one unique response for each prompt.

et al., 2017), CLIP (Radford et al., 2021) and Inception Score (Salimans et al., 2016) for images). Therefore one can confidently use an undetectable watermark without any concern that the quality might degrade. And if the detection key is kept sufficiently private, then the competitive disadvantage to using the watermark can be minimized: One can give the detection key only to mass distributors of information (like Meta and X), so that broad dissemination of AI-generated disinformation can be filtered without interfering in users' personal affairs. Since only the mass information distributors would be able to detect the watermark, it would not harm the value of the content except to bad actors.

**The PRC watermark.** In this paper, we introduce the first undetectable[2] watermarking scheme for image generation models. Our scheme works for latent diffusion models (Rombach et al., 2022), with which we generate watermarked images by progressively de-noising initial watermarked noise within the latent space. The key component in our scheme is a pseudorandom error-correcting code, or pseudorandom code (PRC), a cryptographic object introduced by Christ & Gunn (2024). We therefore refer to our scheme as the **PRC watermark** in this work.

At a high level, a PRC allows us to embed a *cryptographically pseudorandom pattern* that is robustly distributed across the entire latent space, ensuring that the watermark operates at a semantic level. The fact that our watermark is embedded at the semantic level, combined with the PRC's error-correcting properties, makes our watermark highly robust — especially to pixel-level watermark removal attacks (Zhao et al., 2023).

Additionally, since the PRC from Christ & Gunn (2024) can be used to encode and decode messages, we can *robustly embed large messages* within the PRC watermark. While the decoder is somewhat less robust than the detector, the detector can still be effectively used in cases where the decoder fails.

Finally, the PRC watermark is highly flexible, requiring no additional model training or fine-tuning, and can be seamlessly incorporated into existing diffusion model APIs. It allows the user to independently set the message length and a desired upper bound on the false positive rate (FPR) at the time of watermark key generation. The false positive rate is rigorous, rather than empirical: If the user sets the desired upper bound on the false positive rate to $F$, then we prove in Theorem 2 that the false positive rate will not exceed $F$.

**Results.** Experiments on quality and detectability are presented in Section 3.1. We emphasize that undetectability theoretically ensures quality preservation, and our scheme is undetectable by the results of Christ & Gunn (2024). Therefore we perform experiments on quality and detectability only to ensure that our scheme is secure enough with our finite choice of parameters.

We demonstrate the undetectability of our scheme in three key ways:

- We show in Table 1 that the quality, as measured by the FID, CLIP, and Inception Score, are all preserved by the PRC watermark. This is in contrast to **every other scheme we tested**.
- We show in Table 2 that the perceptual variability of responses, as measured by the LPIPS score (Zhang et al., 2019), is preserved under the PRC watermark. This is in contrast to **every other comparable scheme we tested**.
- We show in Figure 1 that an image classifier fails to learn to detect the PRC watermark. The same image classifier quickly learns to detect **every other scheme we tested**.

We demonstrate the robustness of our scheme in Section 3.2. We find that watermark removal attacks fail to remove the PRC watermark without significantly degrading the quality of the image. We test the robustness under ten different types of watermark removal attacks with varying strengths and compare PRC watermark to eight different state-of-the-art watermarking schemes. Among the three watermarking schemes with the lowest impact on image quality,[3] the PRC watermark is the most robust to all attacks.

Finally, we show in Section 3.2 that the PRC watermark can be used to encode and decode long messages in generated images. The encoding algorithm is exactly the same, except that the user passes it a message, and the scheme remains heuristically undetectable. These messages could be

---

[2]See Appendix E.3 for a discussion of the extent to which our scheme is cryptographically undetectable for various choices of parameters.

[3]These are the DwtDct, DwtDctSvd, and PRC watermarks.

used to encode, for instance, timestamps, user IDs, digital signatures, or model specifications. We find in Figure 10 that the robustness of the decoder for 512-bit messages is comparable to, although slightly less than, the robustness of the detector. For non-attacked images, we show in Figure 11 that we can increase the message capacity to at least 2500 bits.

Due to space limitations, the discussion of related work is deferred to Appendix A.

## 2 METHOD

**Threat model.** We consider a setting where users make queries to a provider, and the provider responds to these queries with images produced by some image generation model. In watermarking, the provider holds a *watermarking key* that is used to sample from a modified, *watermarked* distribution over images. Anyone holding the watermarking key can, with high probability, distinguish between samples from the un-watermarked distribution and the watermarked distribution. Since the watermark may be undesirable to some users, some of them may attempt to remove the watermark. We are therefore interested in *robust* watermarks, for which watermark detection still functions even when the image is subjected to a watermark removal attack. We assume that the adversary performing such a removal attack is restricted in two ways. First, the adversary should have weaker capabilities than the provider. If the adversary can generate their own images of high quality, then they don't need to engage in watermark removal attacks. Second, we are only interested in adversaries that produce high-quality images after the removal attack. If removal attacks require significantly degrading the quality of the image, then there is incentive to leave the watermark. We are also interested in *spoofing attacks*, whereby an adversary who doesn't know the watermarking key attempts to add a watermark to an un-watermarked image. We only perform limited experiments on spoofing attacks, so we do not discuss the adversarial capabilities here. However, we note that our techniques, together with the ideas on unforgeable public attribution from Christ & Gunn (2024), immediately yield a scheme that is provably resilient to spoofing attacks.

### 2.1 OVERVIEW OF THE PRC WATERMARK

**Image generation and randomness recovery.** Before describing our watermarking scheme, let us establish some notation. Let Generate be a randomized algorithm that takes as input (1) a prompt string $\boldsymbol{\pi} \in \Sigma^*$ over some alphabet $\Sigma$ and (2) a standard Gaussian in $\mathbb{R}^n$, and produces an output in $\mathbb{R}^d$. Our method applies to any such algorithm, but in this work, we are interested in the case that Generate is a generative image model taking prompts in $\Sigma^*$ and initial (random) latents in $\mathbb{R}^n$ to images in $\mathbb{R}^d$.

Some of the most popular generative image models today are latent diffusion models (Rombach et al., 2022), which consist of a diffusion model specified by a de-noising neural network $\epsilon$, a (possibly-randomized) function $f_\epsilon$ depending on $\epsilon$, a number of diffusion iterations $T$, and an autoencoder $(\mathcal{E}, \mathcal{D})$. For a latent diffusion model, Generate works as follows.

> **Algorithm** Generate$(\boldsymbol{\pi}, \boldsymbol{z}^{(T)})$ :
> (1)    For $i = T$ down to 1:
> (2)          $\boldsymbol{z}^{(i-1)} \leftarrow f_\epsilon(\boldsymbol{\pi}, \boldsymbol{z}^{(i)}, i)$
> (3)    $\boldsymbol{x} \leftarrow \mathcal{D}(\boldsymbol{z}^{(0)})$
> (4)    **Output** $\boldsymbol{x}$

In words, Generate works by starting with a normally distributed latent and iteratively de-noising it. The de-noised latent is then decoded by the autoencoder. In order to produce an image for the prompt $\boldsymbol{\pi}$ using Generate, we use Sample defined as follows.

> **Algorithm** Sample$(\boldsymbol{\pi})$ :
> (1)    Sample $\boldsymbol{z}^{(T)} \sim \mathcal{N}(\boldsymbol{0}, \boldsymbol{I}_n)$
> (2)    Compute $\boldsymbol{x} \leftarrow$ Generate$(\boldsymbol{\pi}, \boldsymbol{z}^{(T)})$
> (3)    **Output** $\boldsymbol{x}$

Detection of the watermark will rely on a separate algorithm, Recover, that recovers an approximation of the latent in $\mathbb{R}^n$ from a given image $\boldsymbol{x} \in \mathbb{R}^d$. For latent diffusion models, the key component in Recover is an *inverse diffusion process* $\delta$ that attempts to invert $\epsilon$ without knowledge of the text prompt $\boldsymbol{\pi}$. There is also some (possibly-randomized) function $g_\delta$ that determines how $\delta$ is used to perform each update.

    **Algorithm** Recover($\boldsymbol{x}$) :
(1)    Compute an initial estimate $\boldsymbol{z}^{(0)} \leftarrow \mathcal{E}(\boldsymbol{x})$ of the de-noised latent.[4]
(2)    For $i = 0$ to $T - 1$:
(3)        $\boldsymbol{z}^{(i+1)} \leftarrow g_\delta(\boldsymbol{z}^{(i)}, i)$
(4)    **Output** $\boldsymbol{z}^{(T)}$

There has been increasing interest in tracing the diffusion model generative process back (Recover). Diffusion inversion has been important for various applications such as image editing (Hertz et al., 2022) and style transfer (Zhang et al., 2023). A commonly used method for reversing the diffusion process is Denoising Diffusion Implicit Models (DDIM) (Song et al., 2021) inversion, which leverages the formulation of the denoising process in diffusion models as an ordinary differential equation (ODE). However, the result of DDIM inversion, $\boldsymbol{z}^{(T)}$, is an approximation even when the input text is known. For our implementation of Generate, we employ Stable Diffusion with DPM-solvers (Lu et al., 2022) for sampling. In our implementation of Recover, we adopt the exact inversion method proposed in Hong et al. (2023) for more accurate inversion.

**Embedding and detecting the watermark.** Our watermarking scheme works by passing to Generate a vector $\tilde{\boldsymbol{z}}^{(T)}$ which is computationally indistinguishable from a sample from $\mathcal{N}(\boldsymbol{0}, \boldsymbol{I}_n)$. To sample $\tilde{\boldsymbol{z}}^{(T)}$, we rely on a cryptographic object called a pseudorandom code (PRC), introduced by Christ & Gunn (2024). A PRC is a keyed error-correction scheme with the property that any polynomial number of encoded messages are jointly indistinguishable from random strings. For watermarking, it suffices to use a *zero-bit* PRC which only encodes the message '1.' If one wishes to encode long messages in the watermark, we can do this as well; see Appendix E for details on how this is accomplished. For simplicity we focus on the zero-bit case in this section.

Our PRC consists of four algorithms, given in Appendix D:

- PRC.KeyGen($n, F, t$) samples a PRC key k, which will also serve as the watermarking key. The parameter $n$ is the block length, which in our case is the dimension of the latent space; $F$ is the desired false positive rate; and $t$ is a parameter which may be increased for improved undetectability at the cost of robustness.

- PRC.Encode$_\mathsf{k}$ samples a PRC codeword.

- PRC.Detect$_\mathsf{k}(\boldsymbol{c})$ tests whether the given string $\boldsymbol{c}$ came from the PRC.

- PRC.Decode$_\mathsf{k}(\boldsymbol{c})$ decodes the message from the given string $\boldsymbol{c}$, if it exists. The decoder is slower and less robust than the detector.

As our PRC, we use the LDPC construction from Christ & Gunn (2024), modified to handle soft decisions. Essentially, this PRC works by sampling random $t$-sparse parity checks and using noisy solutions to the parity checks as PRC codewords. For appropriate choices of parameters, Christ & Gunn (2024) prove that this distribution is cryptographically pseudorandom. We describe how the PRC works in detail in Appendix D, and we describe our watermarking algorithms in detail in Appendix E.

To set up our robust and undetectable watermark, we simply sample a key k using PRC.KeyGen. To sample a watermarked image, we choose $\tilde{\boldsymbol{z}}^{(T)}$ to be a sample from $\mathcal{N}(\boldsymbol{0}, \boldsymbol{I}_n)$ *conditioned on having signs chosen according to the PRC* and then apply Generate. In more detail, we sample $\tilde{\boldsymbol{z}}^{(T)}$ using the following algorithm.

---

[4]In fact, the algorithm of Hong et al. (2023) further uses gradient descent on $\boldsymbol{z}^{(0)}$ to minimize $\| \mathcal{D}(\boldsymbol{z}^{(0)}) - \boldsymbol{x}\|$, initializing with $\boldsymbol{z}^{(0)} = \mathcal{E}(\boldsymbol{x})$. They call this "decoder inversion," and it significantly reduces the recovery error.

**Algorithm** PRCWat.Sample$_k(\pi)$ :

(1)  Sample a PRC codeword $c \in \{-1, 1\}^n$ using PRC.Encode(k)

(2)  Sample $y \sim \mathcal{N}(0, I_n)$

(3)  Let $\tilde{z}^{(T)} \in \mathbb{R}^n$ be the vector defined by $\tilde{z}_i^{(T)} = c_i \cdot |y_i|$ for all $i \in [n]$

(4)  Compute $x \leftarrow$ Generate$(\pi, \tilde{z}^{(T)})$

(5)  **Output** $x$

For a full description of the algorithm, see Algorithm 6.

Since the signs of $z^{(T)} \sim \mathcal{N}(0, I_n)$ are uniformly random, the pseudorandomness of the PRC implies that any polynomial number of samples $\tilde{z}^{(T)}$ in PRCWat.Sample are indistinguishable from samples $z^{(T)} \sim \mathcal{N}(0, I_n)$. As Generate is an efficient algorithm, this yields Theorem 1, which says that our watermarking scheme is undetectable against poly$(n)$-time adversaries for latent space of dimension $n$, as long as the underlying PRC is.

**Theorem 1** (Undetectability). *Let* PRC *be any PRC, and let* PRCWat.Sample *be as defined above. Then for any efficient algorithm $\mathcal{A}$ and any $c > 0$,*

$$\left| \Pr_{k \sim \text{PRC.KeyGen}}[\mathcal{A}^{\text{PRCWat.Sample}_k}(1^n) = 1] - \Pr[\mathcal{A}^{\text{Sample}}(1^n) = 1] \right| \leq \frac{1}{2} + O(n^{-c}).$$

The notation $\mathcal{A}^{\mathcal{O}}(1^n)$ means that $\mathcal{A}$ is allowed to run in any time that is polynomial in $n$, making an arbitrary number of queries to $\mathcal{O}$. For our experiments, we do not strictly adhere to the parameter bounds required for the pseudorandomness proof of Christ & Gunn (2024) to hold; as a result of this and the fact that we use small finite choices of parameters, our scheme should not be used for undetectability-critical applications. See Appendix E.3 for a discussion on this point.

To detect the watermark with the watermarking key, we use (roughly) the following algorithm. As long as Recover reproduces a good enough approximation to the latent that was originally used to generate an image, PRCWat.Detect will recognize the watermark.

**Algorithm** PRCWat.Detect$_k(x)$ :

(1)  Compute $z^{(T)} \leftarrow$ Recover$(x)$

(2)  Let $c$ be the vector of signs of $z^{(T)}$

(3)  Compute result $\leftarrow$ PRC.Detect$_k(c)$

(4)  **Output** result

For our actual detector, we use a slightly more complicated algorithm that accounts for the fact that coordinates of $z^{(T)}$ with larger magnitude are more reliable. The complete algorithm is given in Algorithm 7.

It turns out that, for low error rates, the PRC from Christ & Gunn (2024) can be used to encode and decode long messages using an algorithm called *belief propagation*. We can therefore include long messages in our watermark. Our algorithm for decoding the message from an image is PRCWat.Decode, described in Algorithm 8. PRCWat.Decode is slower and less robust than PRCWat.Detect, but we find that it still achieves an interesting level of robustness.

Finally, our PRC watermark allows the user to set a desired false positive rate, $F$. We prove Theorem 2, which says that our PRC watermark detector has false positive rate at most $F$, in Appendix E.2.

**Theorem 2** (False positive rate). *Let $n, t \in \mathbb{N}$ and $F > 0$. For any image $x$,*

$$\Pr_{k \sim \text{PRCWat.KeyGen}(n,F,t)}[\text{PRCWat.Detect}_k(x) = \text{True}] \leq F$$

*and*

$$\Pr_{k \sim \text{PRCWat.KeyGen}(n,F,t)}[\text{PRCWat.Decode}_k(x) \neq \text{None}] \leq F.$$

In words, Theorem 2 says that any image generated independently of the watermarking key has at most a probability of $F$ of being identified as "watermarked" by our watermark detector or decoder.

Table 1: FID, CLIP and Inception Score (Mean$_{\text{Standard Error}}$) for different watermarks in both COCO and Stable Diffusion Prompts datasets. The PRC watermark is the only one that preserves quality across all three metrics in both datasets.

| Watermark | COCO Dataset | | | Stable Diffusion Prompts Dataset | | |
|---|---|---|---|---|---|---|
| | FID ↓ | CLIP Score ↑ | Inception Score ↑ | FID ↓ | CLIP Score ↑ | Inception Score ↑ |
| Original | $76.3987_{0.3120}$ | $0.4792_{0.0025}$ | $17.5430_{0.1219}$ | $63.4625_{0.2507}$ | $0.6119_{0.0018}$ | $7.4969_{0.0905}$ |
| DwtDct | $76.5676_{0.2237}$ | $0.4761_{0.0022}$ | $17.4686_{0.1228}$ | $63.6912_{0.2588}$ | $0.6047_{0.0018}$ | $7.1346_{0.0951}$ |
| DwtDctSvd | $76.3322_{0.2739}$ | $0.4727_{0.0022}$ | $17.4234_{0.1334}$ | $64.4768_{0.2147}$ | $0.5945_{0.0016}$ | $7.1253_{0.0894}$ |
| RivaGAN | $77.7440_{0.2494}$ | $0.4719_{0.0025}$ | $17.2669_{0.1298}$ | $65.7144_{0.2511}$ | $0.6064_{0.0017}$ | $7.1828_{0.0956}$ |
| StegaStamp | $79.8856_{0.2505}$ | $0.4693_{0.0023}$ | $16.8832_{0.1307}$ | $66.8853_{0.2613}$ | $0.6103_{0.0015}$ | $6.3343_{0.1003}$ |
| SSL | $77.9346_{0.2254}$ | $0.4707_{0.0018}$ | $17.1920_{0.1277}$ | $65.0303_{0.2434}$ | $0.6061_{0.0008}$ | $7.0923_{0.5629}$ |
| Stable Signature | $78.2577_{0.2634}$ | $0.4704_{0.0014}$ | $16.8753_{0.1317}$ | $70.1263_{0.2539}$ | $0.5941_{0.0013}$ | $6.8113_{0.1220}$ |
| Tree-Ring | $77.3445_{0.1733}$ | $0.4795_{0.0035}$ | $17.3989_{0.1399}$ | $68.7192_{0.1572}$ | $0.5964_{0.0013}$ | $7.4173_{0.0940}$ |
| Gaussian Shading | $77.9279_{0.2168}$ | $0.4766_{0.0026}$ | $17.0658_{0.0762}$ | $69.9333_{0.1237}$ | $0.6132_{0.0013}$ | $7.3035_{0.0723}$ |
| PRC | $76.5979_{0.2746}$ | $0.4773_{0.0039}$ | $17.4734_{0.1677}$ | $63.7350_{0.3511}$ | $0.6146_{0.0014}$ | $7.5000_{0.0817}$ |

## 3 EXPERIMENTS

**Experiment overview.** Our experiments focus on text-to-image latent diffusion models, primarily using the Stable Diffusion framework (Rombach et al., 2022). We evaluate various watermarking schemes with `Stable Diffusion-v2.1`[5]. Images are generated at a resolution of $512 \times 512$ with 50 steps using DPMSolver (Lu et al., 2022), applying a classifier-free guidance scale of 3.0. PRC watermarking and VAE models (Kingma & Welling, 2013) are explored in Appendix F, utilizing the inversion method from Hong et al. (2023). All experiments are conducted on NVIDIA H100 GPUs. We compare post-processing methods like DwtDct (Al-Haj, 2007), StegaStamp (Tancik et al., 2020), SSL Watermark (Fernandez et al., 2022), and others, along with in-processing methods like Stable Signature (Fernandez et al., 2023) and Gaussian Shading (Yang et al., 2024). Bit-length varies from 32 to 96, and we encode 512 random bits for PRC. Baseline methods use publicly available code with default parameters. Figure 4 shows visual examples. Watermarking is evaluated on MS-COCO (Lin et al., 2014) and the Stable Diffusion Prompt dataset.[6] We generate 500 images and assess effectiveness (TPR@FPR=0.01), image quality, robustness, and detectability. PRC watermark achieves TPR=1.0@FPR=0.01. For a more detailed description of the experiment setup, please refer to Appendix B.

### 3.1 QUALITY AND DETECTABILITY

To evaluate the image quality of watermarked images, we compute the Frechet Inception Distance (FID) (Heusel et al., 2017), CLIP Score (Radford et al., 2021), and Inception Score (Salimans et al., 2016) to measure the distance between generated watermarked and un-watermarked images, and between watermarked and real images. For our comparison to real images, we use the MS-COCO-2017 training set; for the comparison to un-watermarked images, we use 8,000 images generated by the un-watermarked diffusion model using prompts from the SDP dataset. We calculate FID and CLIP Scores over five-fold cross-validation and report the mean and standard error. To assess perceptual variability (diversity), we select 10 diverse prompts from the PromptHero website[7] and use different in-processing watermark methods to generate 100 images for each prompt. We calculate perceptual similarity for all image pairs using the LPIPS (Zhang et al., 2019) score, averaging the results over the 10 prompts and reporting the standard error. Higher LPIPS scores indicate better variability for a given prompt. This evaluation is essential since, for image generation tasks, users typically generate multiple images from a single prompt and then select the best one (e.g., Midjourney).

Table 2: LPIPS scores (Zhang et al., 2018) for in-processing schemes. Smaller scores mean generated images were more similar according to the LPIPS perceptual metric.

| Watermark | Variability |
|---|---|
| Original | $0.7570_{0.0018}$ |
| Stable Signature | $0.7313_{0.0020}$ |
| Tree-Ring | $0.7413_{0.0021}$ |
| Gaussian Shading | $0.6503_{0.0021}$ |
| PRC | $0.7589_{0.0019}$ |

---

[5]https://huggingface.co/stabilityai/stable-diffusion-2-1-base

[6]https://huggingface.co/datasets/Gustavosta/Stable-Diffusion-Prompts

[7]https://prompthero.com/

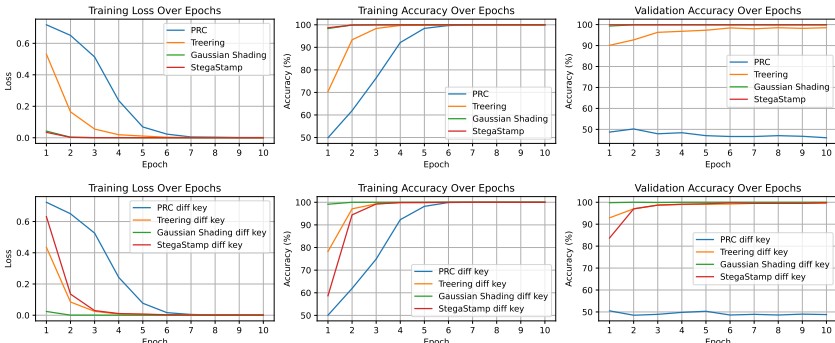

Figure 1: Top: Training a model to detect the watermark without the key. Bottom: Training a model to distinguish between watermarked images generated with different watermarking keys.

Table 1 presents the empirical results for FID, CLIP, and Inception Scores for different watermarking schemes on both the COCO and SDP datasets. The table compares original image quality, post-processing watermark quality, and in-processing watermark quality (separated by dashed lines). We observe that StegaStamp results in the most significant quality degradation among post-processing watermark schemes, and the PRC watermark is the only method that consistently preserves image quality across all three metrics on both datasets. Table 2 presents the results of the variability analysis. Since post-processing methods are expected to have minimal impact on image variability, they are excluded from this table. The PRC watermark demonstrates variability comparable to un-watermarked images, outperforming the other in-processing schemes in this regard.

To evaluate detectability, we use ResNet18 (He et al., 2016) as the backbone model and train it on 7,500 un-watermarked images and 7,500 watermarked images (or 7,500 images watermarked with key 1 and 7,500 with key 2) to perform binary classification. Each experiment tests different watermarking schemes, with results shown in Figure 1. For the PRC watermark, the neural network slowly converges to perfect detection on the training set but achieves only 50% accuracy (random guess) on the validation set, indicating that the network is memorizing the training samples rather than learning the watermark pattern. In contrast, for all other schemes, the network performs perfectly on the validation set, demonstrating that the watermark is learnable.

## 3.2 ROBUSTNESS OF THE DETECTOR

To comprehensively evaluate the robustness of the PRC watermark and compare it to baseline watermarking methods, we tested nine distinct watermarking techniques against ten different types of attacks. Detailed descriptions of the attack configurations can be found in Appendix C.1.

The robustness of the various watermarking methods under these attacks is shown in Figure 5. We evaluated the quality of the attacked images using PSNR, SSIM, and FID metrics, comparing them to the original watermarked images. Notably, the PRC watermark demonstrates high resilience to most attacks. Even under sophisticated attacks, no method successfully reduced the true positive rate (TPR) below 0.99 while keeping the FID score under 70. This demonstrates that current watermark removal techniques struggle to erase our watermark without significantly degrading image quality. For instance, as shown in Figure 5, a JPEG compression attack with a quality factor of 20 only reduced the TPR from 1.0 to 0.94, but the resulting images displayed noticeable blurriness and a loss of detail (see Figure 3). Finally, in Figure 7 we demonstrate increased robustness for $t = 2$.[8] However, for $t = 2$ there exist fast attacks on the undetectability of the PRC watermark, so we do not explore this choice further.

**Encoding long messages in the watermark.** The use of a PRC allows us to embed long messages in our watermarks, as described in Appendix E. We find in Figure 10 that the decoder is highly robust for 512-bit messages, although the detector is slightly more robust in this case. We find in Figure 11 that the decoder can reliably recover up to 2500 bits of information if the images are not subjected to removal attacks.

---

[8]For our other experiments we set $t = 3$. See Appendix D for details on the meaning of the parameter $t$.

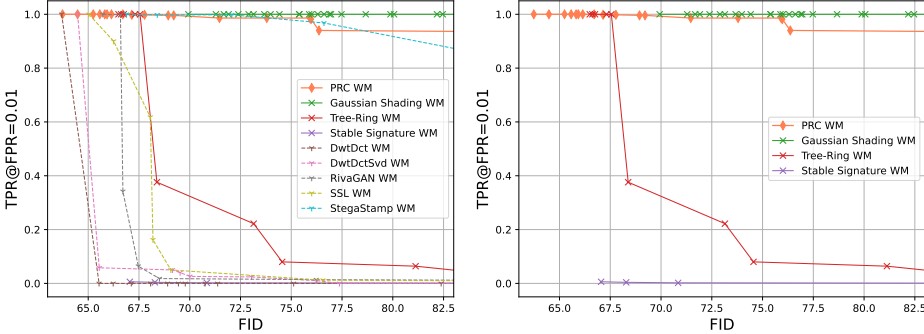

Figure 2: Robustness under the strongest attacks, excluding the embedding attack. We show all points from the corresponding plot in Figure 5 for which there is no other point with a higher FID and TPR. In the figure on the right, we only include the in-processing watermarks. The TPR for the PRC watermark drops after the FID reaches 75; this corresponds to the JPEG 20 attack, of which we give an example in Figure 3.

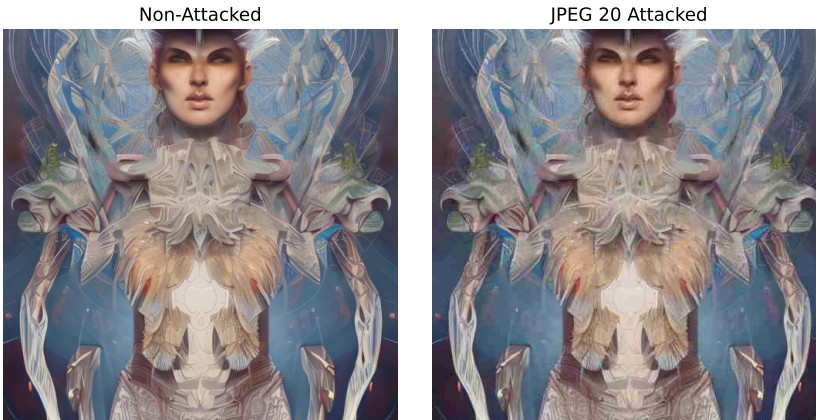

Figure 3: Example images under the JPEG 20 attack with a PSNR of 28.39. Notice the blurriness and lack of detail in the attacked image.

**Security of the PRC watermark under spoofing attacks.** To test the spoofing robustness of different watermarks, we followed the approach in Saberi et al. (2023), aiming to classify non-watermarked images as watermarked (increasing the false positive rate). Spoofing attacks can damage the reputation of generative model developers by falsely attributing watermarks to images. We used a PGD-based (Madry et al., 2018) method similar to that of the surrogate model adversarial attacks, flipping the surrogate model's prediction from un-watermarked to watermarked. Just as with the adversarial surrogate attack, this attack cannot work against any undetectable watermark such as PRC watermark.

**Possibility of extension.** The PRC watermark can also be applied to other generative models, particularly those sampling from Gaussian distributions. We have set up a demo experiment working for traditional VAE models, as detailed in Appendix F. We would also be interested to see the PRC watermark applied to emerging generative models such as Flow matching (Lipman et al., 2022); whether or not this is possible hinges only on the existence of a suitable Recover algorithm.

## 4 CONCLUSION

We give a new approach to watermarking for generative image models that incurs no observable shift in the generated image distribution and encodes long messages. We show that these strong guarantees do not preclude strong robustness: Our watermarks achieve robustness that is competitive with state-of-the-art schemes that incur large, observable shifts in the generated image distribution.

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

## A    RELATED WORK

There is a rich history of digital watermarking techniques, ranging from conventional steganography to modern methods based on generative models. Following the taxonomy in An et al. (2024), watermarking methods are categorized into two types: post-processing and in-processing schemes.

- **Post-processing schemes** embed the watermark after image generation and have been used for decades due to their broad applicability.
- **In-processing schemes** modify the generative model or sampling process to embed the watermark directly in the generated content.

Our PRC watermark falls under the in-processing category. Note that post-processing watermarks cannot be made undetectable without introducing extra modeling assumptions: One can always distinguish between a fixed image and any modification of it. We refer the reader to surveys (Cox et al., 2008; Wan et al., 2022; An et al., 2024) for more on post-processing methods. Below, we focus on two popular in-processing techniques: Tree-Ring and Gaussian Shading watermarks.

**Tree-Ring watermark.**    Wen et al. (2023) introduced Tree-Ring watermark, the first in-processing watermark that modifies the latent sampling distribution and employs an inverse diffusion process for detection. Our PRC watermark builds on this framework but adopts a different latent distribution. The Tree-Ring watermark works by fixing concentric rings in the Fourier domain of the latent space to be 0. To detect the watermark, one uses DDIM inversion (Song et al., 2021) to estimate the initial latent, and the watermark is considered present if the latent estimate has unusually small values in the watermarked rings. Follow-up works have extended this approach by refining the heuristic latent pattern in the watermarking process (Zhang et al., 2024; Ci et al., 2024). However, under Tree-Ring's strategy, the initial latent significantly deviates from the Gaussian latent distribution, leading to reduced image quality and variability, as shown in Tables 1 and 2. Furthermore, the Tree-Ring watermark is a zero-bit scheme and cannot encode messages. While the Tree-Ring watermark is robust to several attacks, it is highly susceptible to the adversarial surrogate attack since the latent pattern is easy to learn with a neural network. In Figure 5, we find that this attack removes the Tree-Ring watermark with minimal effect on image quality.

**Gaussian Shading watermark.**    The basic Gaussian Shading watermark (Yang et al., 2024) works by choosing a fixed quadrant of latent space as the watermarking key, and only generating images from latents in that quadrant. Detection involves recovering the latent and determining if it lies unusually close to the watermarked quadrant. In their paper, Yang et al. (2024) include a proof that Gaussian Shading has "lossless performance." However, this proof only shows that the distribution of a *single* watermarked image is the same as that of a single un-watermarked image. Crucially, even standard quality metrics such as the FID (Heusel et al., 2017), CLIP Score (Radford et al., 2021), and Inception Score (Salimans et al., 2016) account for correlations between generated images, so their proof of lossless performance *does not guarantee perfect quality under these metrics*. Indeed, we find in Table 1 that the Gaussian Shading watermark significantly degrades the FID and Inception Score.[9] We expand on this by measuring the "variability" of watermarked images. Since images under the Gaussian Shading watermark all come from the same quadrant in latent space, we expect that the variability should be reduced. We use the LPIPS perceptual similarity score (Zhang et al., 2018) to measure the diversity among different watermarked images for a fixed prompt. As shown in Table 2, the perceptual similarity between images is significantly higher with the Gaussian Shading watermark, confirming the diminished variability.

**Undetectability.**    Undetectable watermarks were initially defined by Christ et al. (2024) in the context of language models. Subsequent to Christ & Gunn (2024), alternative constructions of PRCs have been given by Golowich & Moitra (2024) and Ghentiyala & Guruswami (2024). It would be interesting to see if these PRCs yield improved image watermarks, but we did not investigate this.

---

[9]Table 1 of Yang et al. (2024) appears to show that the FID against the COCO dataset is preserved under Gaussian Shading. However, from their code repository it appears that this table is generated by *re-sampling the watermarking key for every generation*. To be consistent with the intended use case, in this work we use the same random watermarking key to generate many images and compute the quality score.

## B    More on experiment setup

In our primary experiments, we focus on text-to-image latent diffusion models, utilizing the widely adopted Stable Diffusion framework (Rombach et al., 2022). Specifically, we evaluate the performance of various watermarking schemes using the `Stable Diffusion-v2.1`[10] model, a state-of-the-art generative model for high-fidelity image generation. Additionally, we explore applying PRC watermarking to other generative models, as demonstrated with VAE (Kingma & Welling, 2013) models in Appendix F. All images are generated at a resolution of 512×512 with a latent space of 4×64×64. During inference, we apply a classifier-free guidance scale of 3.0 and sample over 50 steps using DPMSolver (Lu et al., 2022). As described in Section 2, we perform diffusion inversion using the exact inversion method from Hong et al. (2023) to obtain the latent variable $z^{(T)}$. In particular, we use 50 inversion steps and an inverse order of 0 to expedite detection, balancing accuracy and computational efficiency. All experiments are conducted on NVIDIA H100 GPUs.

**Watermark baselines.**    We conduct comparative evaluations against various watermarking schemes, including in-, and post-processing techniques, as defined in Appendix A. For post-processing methods, we compare with DwtDct (Al-Haj, 2007), DwtDctSvd (Navas et al., 2008), RivaGAN (Zhang et al., 2019), StegaStamp (Tancik et al., 2020), and SSL Watermark (Fernandez et al., 2022). For in-processing methods, we include a comparison with Stable Signature (Fernandez et al., 2023), Tree-Ring (Wen et al., 2023) and Gaussian Shading (Yang et al., 2024). Most baseline methods are designed to embed multi-bit strings within an image. Specifically, we set 32 bits for DwtDctSvd, RivaGAN, and SSL Watermark; 96 bits for StegaStamp; and 48 bits for Stable Signature. We employ publicly available code for each method, using the default inference and fine-tuning parameters specified in original respective papers for post- and in-processing methods. For Tree-Ring and Gaussian Shading watermarks, we use the same diffusion model and inference parameter settings as those used in PRC. We encode 512 random bits in the PRC watermark. If the decoder is successful, then with high probability, the bits are recovered correctly. Figure 4 illustrates examples of different watermarking schemes applied to a specific text prompt, highlighting the visual impact of each approach.

**Datasets and evaluation.**    We evaluate watermarking methods on two datasets: MS-COCO (Lin et al., 2014) and the Stable Diffusion Prompt (SDP) dataset.[11] We generate 500 un-watermarked images using MS-COCO captions or SDP prompts, and apply post-processing watermark methods to generate watermarked images. In-processing methods directly generate watermarked images from prompts. To assess the performance of the different watermarking schemes, we primarily examine four aspects: effectiveness, image quality, robustness, and detectability. For effectiveness, which involves performing binary classification between watermarked and un-watermarked images, we calculate the true positive rate (TPR) at a fixed false positive rate (FPR). Specifically, we report TPR@FPR=0.01. Without any attacks, the PRC watermark achieves TPR=1.0@FPR=0.01. Note that for PRC watermarking, the FPR is set at 1%, though it can be easily made smaller depending on the use case (see long message experiments in Section 3.2).

## C    Additional experiment results and details on robustness

### C.1    Additional experiment results

The figures included in this section are:

- Figure 4, examples of different watermarks applied to one text prompt.
- Figure 5, a comprehensive evaluation of watermarking schemes under the attacks described in Appendix C.2.
- Figure 6, the performance of the embedding attack on in-processing watermarks.
- Figure 7, a brief evaluation of the robustness of our PRC watermark with $t = 2$.
- Figure 8, the performance of the spoofing attack against in-processing watermarks.

---

[10] https://huggingface.co/stabilityai/stable-diffusion-2-1-base
[11] https://huggingface.co/datasets/Gustavosta/Stable-Diffusion-Prompts

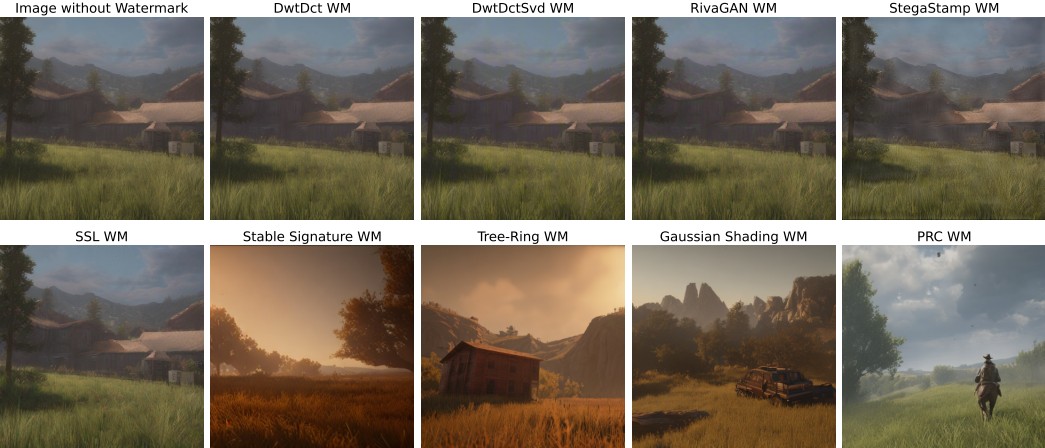

Figure 4: Examples of different watermarks applied to the image generated with the prompt: *"red dead redemption 2, cinematic view, epic sky, detailed, concept art, low angle, high detail, warm lighting, volumetric, godrays, vivid, beautiful, trending on artstation, by jordan grimmer, huge scene, grass, art greg rutkowski"*. For post-processing watermark methods, the watermarks directly perturb the un-watermarked image. Notably, the StegaStamp watermark introduces visible blurring artifacts.

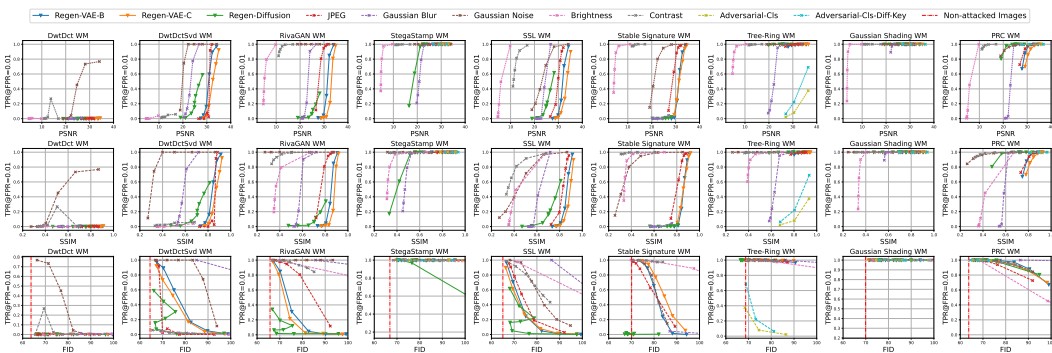

Figure 5: Robustness of various watermarking schemes. PSNR and SSIM are used to measure the similarity between a *single* original image and attacked image. FID is used to measure distance between the *distribution* of un-watermarked images and attacked images. The vertical dotted red line in the FID plots is the FID for un-perturbed watermarked images. Note that the strange behavior of the FID for certain watermarks under the Regen-Diffusion attack can be explained by the attack simply correcting its own errors.

- Figure 9, example images under the embedding attack.
- Figure 10, a brief evaluation of the robustness of our PRC watermark decoder for 512 bits.
- Figure 11, the length of messages which can be reliably encoded and decoded with out PRC watermark when there is no watermark removal attack.

## C.2 DETAILS ON ROBUSTNESS

We applied a range of attacks, categorized into photometric distortions, degradation distortions, regeneration attacks, adversarial attacks, and spoofing attacks. Each type is described in detail below.

**Photometric distortions.** We applied two photometric distortion attacks: brightness and contrast adjustments. For brightness, we tested enhancement factors of $[2, 4, 6, 8, 12]$, where a factor of 0.0 results in a completely black image, and 1.0 retains the original image. Similarly, for contrast, we

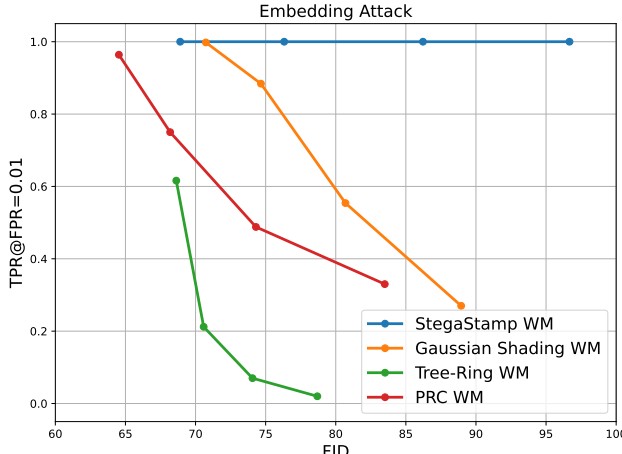

Figure 6: Embedding attack for different watermarks. Only the PRC watermark can attain FID below a certain threshold. Above this threshold, StegaStamp is the strongest scheme we tested against the embedding attack. The embedding attack is quite powerful, as it assumes the attacker knows the VAE used in the diffusion model for embedding latents. However, its effectiveness could be mitigated by employing an adversarially robust VAE encoder or keeping the VAE component of the diffusion model private.

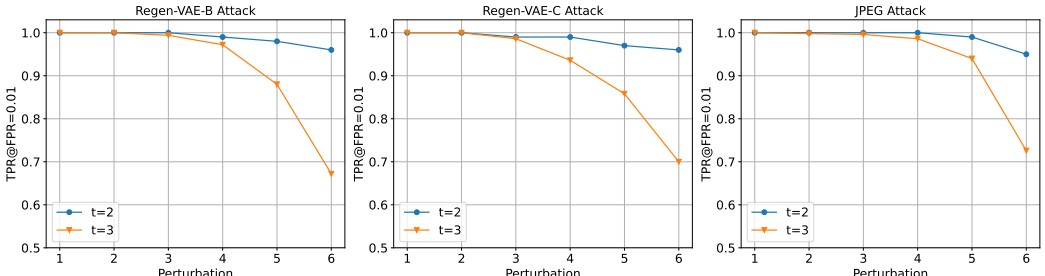

Figure 7: We observe improved robustness for $t = 2$.

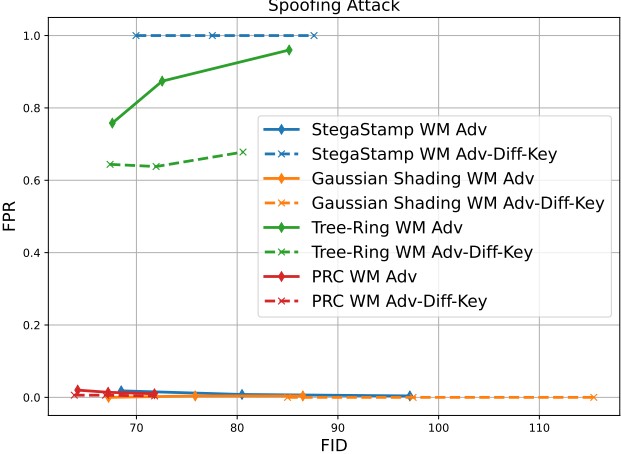

Figure 8: Spoofing attack results. Tree-Ring and StegaStamp are vulnerable to spoofing attacks. Even with the target FPR set to 0.01, adversaries can significantly raise the FPR, causing the watermark detector to misclassify unwatermarked images as watermarked, which can damage the watermark owner's reputation. The spoofing attack does not affect undetectable watermarks like the PRC watermark.

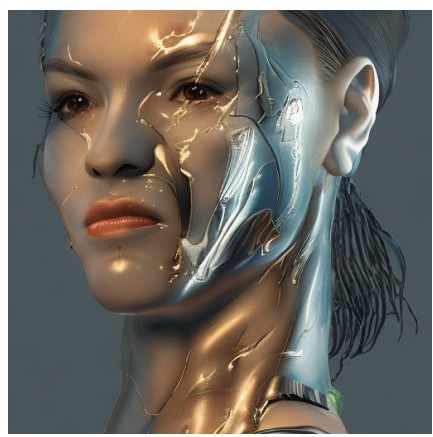

(a) Original image

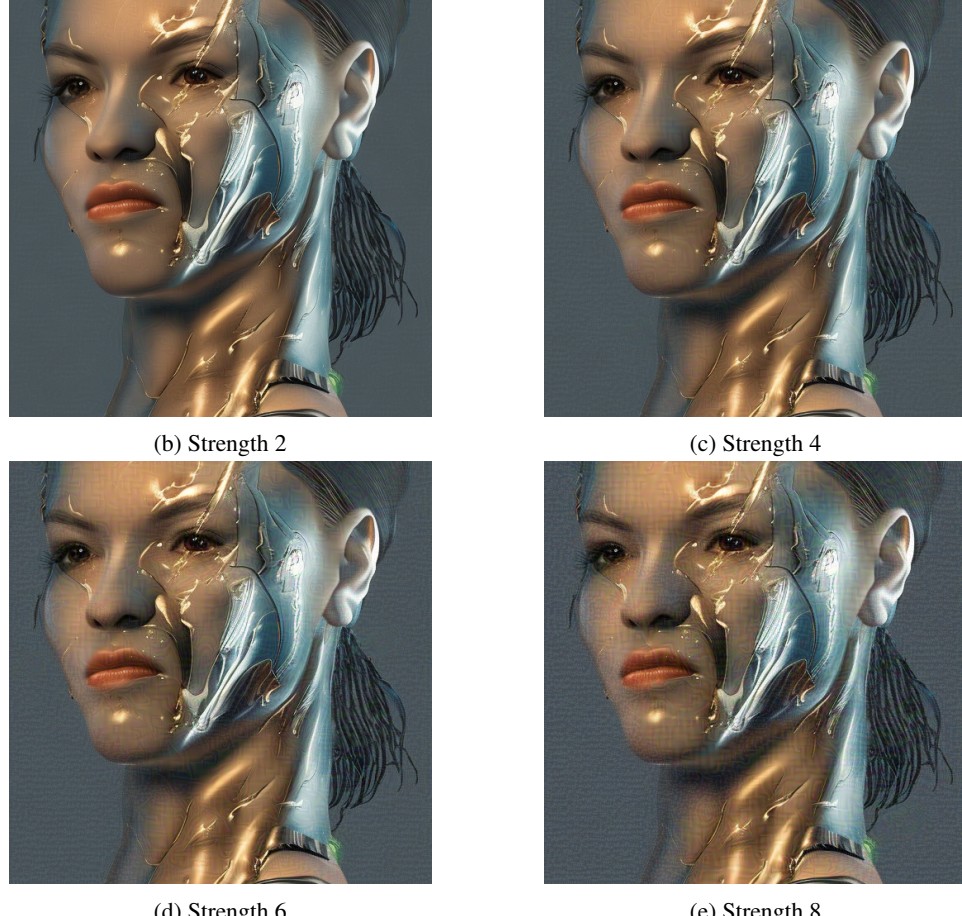

(b) Strength 2                                           (c) Strength 4

(d) Strength 6                                           (e) Strength 8

Figure 9: Example images under the embedding attack. Even the strength-2 embedding attack, for which the PRC attains a detection rate of over 95%, noticeably deteriorates the image quality on close inspection.

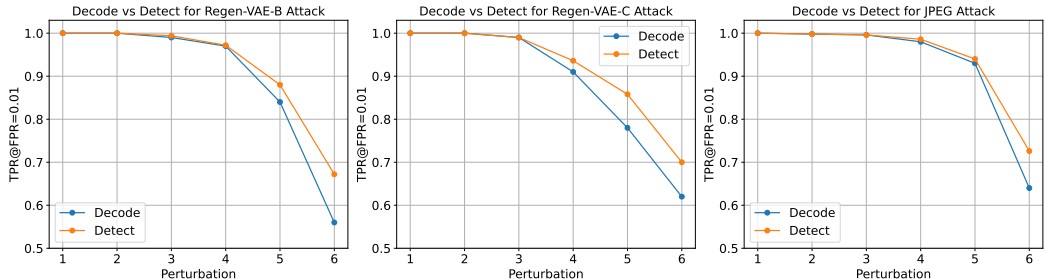

Figure 10: Comparison between robustness of the decoder for 512 bits and the detector. The detector is faster and consistently more robust than the decoder, but the detector does not recover messages in the watermark. The TPR for the decoder is the rate at which the message is correctly decoded.

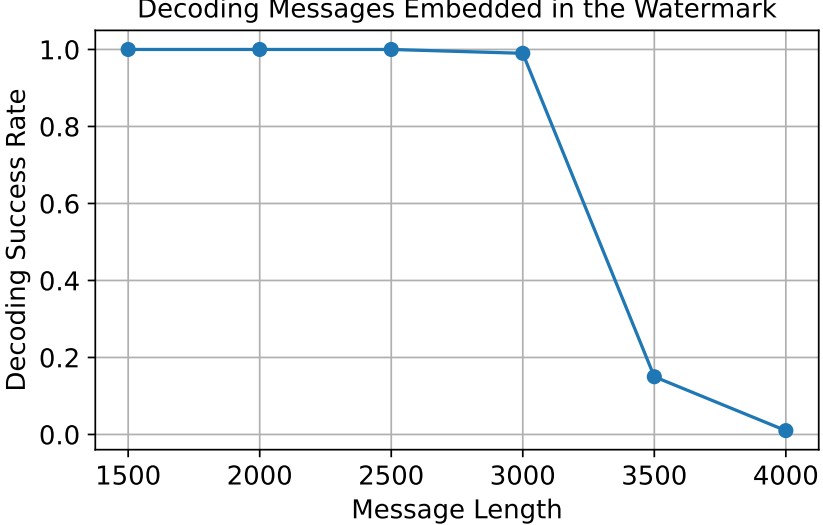

Figure 11: Testing the decoder for large message lengths with no adversarial perturbations. The PRC watermark parameters we used for this experiment are $t = 4$, $F = 1\text{e-}9$, and $\sigma = 0$.

used enhancement factors of $[2, 3, 4, 5, 6, 7]$, where a factor of 0.0 produces a solid gray image, and 1.0 preserves the original image.

**Degradation distortions.** Three types of degradation distortions were applied: Gaussian blur, Gaussian noise, and JPEG compression. Specifically:

- **Gaussian Blur:** We varied the radius from $[2, 4, 6, 8, 10, 12]$.
- **Gaussian Noise:** Noise was introduced with a mean of 0 and standard deviations of $[5, 10, 15, 20, 25, 30]$.
- **JPEG Compression:** Compression quality was set at $[10, 20, 30, 40, 50, 60]$, with lower quality levels leading to higher degradation.

**Regeneration attacks.** Regeneration attacks (Zhao et al., 2023) alter an image's latent representation by first introducing noise and then applying a denoising process. We implemented two forms of regeneration attacks: diffusion model-based and VAE-based approaches.

- **Diffusion Model Regeneration:** We employed the Stable-Diffusion-2-1base model as the backbone and conducted $[10, 20, 30, 50, 80, 100, 150, 200]$ diffusion steps to attack the image. As the number of steps increased, the image diverged further from the original, often causing a performance drop. Interestingly, for the FID metric, we observed that more diffusion steps sometimes improved the FID score, as the diffusion model's inherent purification process preserved a natural appearance while altering textures and styles.
- **VAE-Based Regeneration:** We used two pre-trained image compression models from the CompressAI library[12]: Bmshj2018 Ballé et al. (2018) and Cheng2020 Cheng et al. (2020), referred to as Regen-VAE-B and Regen-VAE-C, respectively. Compression factors were set to $[1, 2, 3, 4, 5, 6]$, where lower compression factors resulted in more heavily degraded images.

**Adversarial attacks.** We also explored adversarial attacks, focusing on surrogate detector-based and embedding-based adversarial methods.

- **Surrogate Detector Attacks:** Following (Saberi et al., 2023), we trained a ResNet18 model (He et al., 2016) on watermarked and non-watermarked images to act as a surrogate classifier. Specifically, we train the model for 10 epochs with a batch size of 128 and a learning rate of 1e-4. Using this model, we applied Projected Gradient Descent (PGD) adversarial attacks (Madry et al., 2018) on test images, simulating an adversary who knows either un-watermarked images and watermarked images (Adversarial-Cls), or watermarked images with two different keys (Adversarial-Cls-Diff-Key). The goal was to perturb the images with one key such that the detector misclassifies them as being associated with the other key. The attack was tested on four watermarking methods: Tree-Ring, Gaussian Shading, PRC, and StegaStamp watermark, with epsilon values of $[4, 8, 12]$. Since the PRC watermark is undetectable, we find in Figure 1 that the surrogate classifier cannot even be trained!
- **Embedding-Based Adversarial Attacks:** Adversarial perturbations were also applied to the image embedding space. Given an encoder $f : \mathcal{X} \rightarrow \mathcal{Z}$ that maps images to latent features, we crafted adversarial images $x_{\text{adv}}$ to diverge from the original watermarked image $x$, constrained within an $l_\infty$ perturbation limit. This was solved using the PGD algorithm (Madry et al., 2018). The VAE model for the original diffusion model `stabilityai/sd-vae-ft-mse` was assumed to be known for this attack.

## D  THE PSEUDORANDOM CODE

We use the construction of a PRC from Christ & Gunn (2024), which is secure under the certain-subexponential hardness of LPN. The proof of pseudorandomness, assuming the $2^{\omega(\sqrt{\lambda})}$ hardness of LPN, from the technical overview of Christ & Gunn (2024) applies identically here. The PRC works by essentially embedding random parity checks in codewords. The key generation and encoding algorithms are given in Algorithms 1 and 2.

---

[12] https://github.com/InterDigitalInc/CompressAI

---

**Algorithm 1:** PRC.KeyGen

---

**Input:** $n$, message_length, $F$, $t$
**Output:** PRC key k

    /* Set parameters                                                     */

1  $\lambda \leftarrow \lfloor \log_2 \binom{n}{t} \rfloor$;

2  $\eta \leftarrow 1 - 2^{-1/\lambda}$;

3  num_test_bits $\leftarrow \lceil \log_2(1/F) \rceil$;

4  $k \leftarrow$ message_length $+ \lambda +$ num_test_bits;

5  $r \leftarrow n - k - \lambda$;

6  max_bp_iter $\leftarrow \lfloor \log_t n \rfloor$;

    /* Sample randomness to ensure a low false-positive rate    */

7  Sample uniformly random vectors otp $\in \mathbb{F}_2^n$ and testbits $\in (\mathbb{F}_2)^{\text{num\_test\_bits}}$;

    /* Sample generator matrix and parity-check matrix        */

8  Sample a uniformly random matrix $\boldsymbol{G}_0 \in \mathbb{F}_2^{(n-r) \times \lambda}$;

9  **for** $i \in \{1, \ldots, r\}$ **do**

10     Sample a random $(t-1)$-sparse vector $\boldsymbol{w}_i \in \mathbb{F}_2^{n-r+i-1}$;

11     $\boldsymbol{G}_i \leftarrow \begin{bmatrix} \boldsymbol{G}_{i-1} \\ \boldsymbol{w}_i^T \boldsymbol{G}_0 \end{bmatrix}$;

12     $\boldsymbol{w}_i' \leftarrow [\boldsymbol{w}_i^T, 1, 0^{r-i}]$;

13  Let $\boldsymbol{P}$ be the matrix whose rows are $\boldsymbol{w}_1', \ldots, \boldsymbol{w}_r'$ and let $\boldsymbol{G} \leftarrow \boldsymbol{G}_r$;

14  Sample a random permutation $\boldsymbol{\Pi} \in \mathbb{F}_2^{n \times n}$ and let $P \leftarrow \boldsymbol{P}\boldsymbol{\Pi}^{-1}, G \leftarrow \boldsymbol{\Pi}\boldsymbol{G}$;

15  k $\leftarrow (n, \text{message\_length}, F, t, \lambda, \eta, \text{num\_test\_bits}, k, r, \text{max\_bp\_iter}, \text{otp}, \text{testbits}, \boldsymbol{G}, \boldsymbol{P})$;

16  Output k;

---

**Algorithm 2:** PRC.Encode

---

**Input:** k, $\boldsymbol{m}$
**Output:** PRC codeword $\boldsymbol{c}$

1  Parse
    $(n, \text{message\_length}, F, t, \lambda, \eta, \text{num\_test\_bits}, k, r, \text{max\_bp\_iter}, \text{otp}, \text{testbits}, \boldsymbol{G}, \boldsymbol{P}) \leftarrow$ k;

2  Sample a uniformly random vector $\boldsymbol{r} \in \mathbb{F}_2^\lambda$;

3  $\boldsymbol{y} \leftarrow (\text{testbits}, \boldsymbol{r}, \boldsymbol{m})$;

4  Sample $\boldsymbol{e} \sim \text{Ber}(n, \eta)$;

5  $\boldsymbol{c} \leftarrow \boldsymbol{G}\boldsymbol{y} \oplus e \oplus \text{otp}$;

6  Output $\boldsymbol{c}$;

---

Since the work of Christ & Gunn (2024), at least two new constructions of PRCs have been introduced using different assumptions (Golowich & Moitra (2024); Ghentiyala & Guruswami (2024)). It would be interesting to see if any of these new constructions yield image watermarks with improved robustness.

The main difference between the PRC used here and the one from the technical overview of Christ & Gunn (2024) is that ours is optimized for our setting by allowing soft decisions on the recovered bits. That is, PRC.Detect takes in not a bit-string but a vector $\boldsymbol{s}$ of values in the interval $[-1, 1]$. If the PRC codeword is $\boldsymbol{c}$, then $s_i$ should be the expected value $(-1)^{c_i}$ conditioned on the user's observation. We present PRC.Detect in Algorithm 3 and explain how we designed it in Appendix E.1.

Christ & Gunn (2024) show that any zero-bit PRC (i.e., a PRC with a Detect algorithm but no Decode) can be generically converted to one that encodes information at a linear rate. However, that construction requires increasing the block-length of the PRC, which could harm the practical performance of our watermark. Instead, we use belief propagation with ordered statistics decoding to directly decode the message. Note that belief propagation cannot handle a constant rate of errors if the sparsity is greater than a constant; therefore, this only works when Recover produces an accurate approximation to the initial latent. Still, since our robustness experiments use a small

---

**Algorithm 3:** PRC.Detect

---

**Input:** k, $\boldsymbol{s}$
**Output:** Detection result True or False

1 Parse
$(n, \text{message\_length}, F, t, \lambda, \eta, \text{num\_test\_bits}, k, r, \text{max\_bp\_iter}, \text{otp}, \text{testbits}, \boldsymbol{G}, \boldsymbol{P}) \leftarrow$ k;

2 For $i \in [n]$, let $s_i \leftarrow (-1)^{\text{otp}_i} \cdot (1 - 2\eta) \cdot s_i$;

3 For each parity check $\boldsymbol{w} \in \boldsymbol{P}$, let $\hat{s}_{\boldsymbol{w}} \leftarrow \prod_{i \in \boldsymbol{w}} s_i$;

4 $C \leftarrow \frac{1}{2} \sum_{\boldsymbol{w} \in \boldsymbol{P}} \log^2 \left( \frac{1 + \hat{s}_i}{1 - \hat{s}_i} \right)$;

5 if

$$\sum_{\boldsymbol{w} \in \boldsymbol{P}} \log \left( \frac{1 + \hat{s}_{\boldsymbol{w}}}{2} \right) \geq \sqrt{C \log(1/F)} + \frac{1}{2} \sum_{\boldsymbol{w} \in \boldsymbol{P}} \log \left( \frac{1 - \hat{s}_{\boldsymbol{w}}^2}{4} \right)$$

then

6 $\quad$ Output True;

7 else

8 $\quad$ Output False;

---

sparsity of $t = 3$, we find that our decoder functions even when the image is subjected to significant perturbation.

---

**Algorithm 4:** PRC.Decode

---

**Input:** k, $\boldsymbol{s}$
**Output:** Decoded message $\boldsymbol{m} \in \{0, 1\}^k$ or None

1 Parse
$(n, \text{message\_length}, F, t, \lambda, \eta, \text{num\_test\_bits}, k, r, \text{max\_bp\_iter}, \text{otp}, \text{testbits}, \boldsymbol{G}, \boldsymbol{P}) \leftarrow$ k;

2 For $i \in [n]$, let $s_i \leftarrow (-1)^{\text{otp}_i} \cdot (1 - 2\eta) \cdot s_i$;

3 $\boldsymbol{y} \leftarrow \text{BP-OSD}(\boldsymbol{G}, \boldsymbol{P}, \boldsymbol{s})$;

4 Parse $(\text{testbits}', \boldsymbol{r}, \boldsymbol{m}) \leftarrow \boldsymbol{y}$;

5 if $\text{testbits}' = \text{testbits}$ then

6 $\quad$ Output $\boldsymbol{m}$;

7 else

8 $\quad$ Output None;

---

The only parameters that need to be set in PRC.KeyGen are:

- $n$, the block length, which is the dimension of the image latents in the PRC watermark. Holding the other parameters constant, larger $n$ will yield a more robust PRC.

- $\text{message\_length}$, the length of messages that can be encoded by PRC.Encode. Increasing $\text{message\_length}$ yields a less robust PRC.

- $F$, the desired false positive rate. We prove in Theorem 2 that the scheme will always have a false positive rate of at most $F$, as long as the string being tested does not depend on the PRC key.

- $t$, the sparsity of parity checks. Larger $t$ yields undetectability against more-powerful adversaries, but decreased robustness.

For watermark detection and decoding, we allow the user to set an estimated error $\sigma$. This should be the standard deviation of the error $\boldsymbol{z}' - \boldsymbol{z}$ that the user expects. In cases where the watermark does not need to be robust to perturbations of the image, one can set $\sigma = 0$. If $\sigma$ is not set by the user, we use a default of $\sigma = \sqrt{3/2}$ which we found to be effective for robust watermarking.

We use the Galois package of Hostetter (2020) for conveniently handling linear algebra over $\mathbb{F}_2$. We use the belief propagation implementation of Roffe (2022) to decode messages in the watermark.

# E    DETAILS ON THE PRC WATERMARK

Watermark key generation, Algorithm 5, is exactly the same as PRC key generation.

---

**Algorithm 5:** PRCWat.KeyGen

---

**Input:** $n$, message_length, $F$, $t$
**Output:** Watermarking key k
1  k $\leftarrow$ PRC.KeyGen($n$, message_length, $F$, $t$);
2  ($n$, message_length, $F$, $t$, $\lambda$, $\eta$, num_test_bits, $k$, $r$, max_bp_iter, otp, testbits, $\boldsymbol{G}$, $\boldsymbol{P}$) $\leftarrow$ k;
3  Output k;

---

Watermarked image generation works by sampling the initial latents to have signs chosen according to a PRC codeword. If a message is to be encoded in the watermark, the message is simply encoded into the PRC.

---

**Algorithm 6:** PRCWat.Sample

---

**Input:** Watermarking key k and message $\boldsymbol{m}$
**Output:** Generated image $\boldsymbol{x}$
1  $\boldsymbol{c} \leftarrow$ PRC.Encode(k, $\boldsymbol{m}$);
2  Sample $\tilde{\boldsymbol{z}} \sim \mathcal{N}(\boldsymbol{0}, \boldsymbol{I}_n)$;
3  **for** $i \in [n]$ **do**
4  $\quad \lfloor \ \tilde{z}_i \leftarrow c_i \cdot |\tilde{z}_i|$;
5  $\boldsymbol{x} \leftarrow$ Generate($\boldsymbol{\pi}, \tilde{\boldsymbol{z}}$);
6  Output $\boldsymbol{x}$;

---

Our detection algorithm PRC.Detect is given in Algorithm 3. In Appendix E.1 we will explain how we designed the detector, and in Appendix E.2 we will prove Theorem 2 which says that PRC.Detect and PRC.Decode have false positive rates of at most $F$. Note that PRC.Decode is guaranteed to have a false positive rate of at most $F$ simply because of testbits.

---

**Algorithm 7:** PRCWat.Detect

---

**Input:** Watermarking key k, image $\boldsymbol{x}$, and estimated error $\sigma$
**Output:** Detection result True or False
1  $\boldsymbol{z} \leftarrow$ Recover($\boldsymbol{x}$);
2  **for** $i \in [n]$ **do**
3  $\quad \lfloor \ s_i = \mathrm{erf}\left( \frac{z_i}{\sqrt{2\sigma^2(1+\sigma^2)}} \right)$;
4  result $\leftarrow$ PRC.Detect(k, $\boldsymbol{s}$);
5  Output result;

---

## E.1    DESIGNING THE WATERMARK DETECTOR

Let $\boldsymbol{z}$ be the initial latent and $\boldsymbol{z}'$ be the recovered latent. We will compute the probability that a given parity check $w$ is satisfied by $\mathrm{sign}(\boldsymbol{z})$ (after accounting for the noise and one-time pad), conditioned on the observation of $\boldsymbol{z}'$. In order for this to be possible, we need to model the distributions of $\boldsymbol{z}$ and $\boldsymbol{z}'$: We use $\boldsymbol{z} \sim \mathcal{N}(\boldsymbol{0}, \boldsymbol{I}_n)$ and $\boldsymbol{z}' \sim \mathcal{N}(\boldsymbol{z}, \sigma^2 \boldsymbol{I}_n)$ for some $\sigma > 0$.

Crucially, when we bound the false positive rate in Appendix E.2, we will do it in a way that *does not depend* on the distribution of $\boldsymbol{z}'$; we only use the facts that $\boldsymbol{z} \sim \mathcal{N}(\boldsymbol{0}, \boldsymbol{I}_n)$ and $\boldsymbol{z}' \sim \mathcal{N}(\boldsymbol{z}, \sigma^2 \boldsymbol{I}_n)$ to inform the design of our detector. In other words, Theorem 2 holds *unconditionally*, even though our detector is designed to have the highest true positive rate for a particular distribution of $\boldsymbol{z}'$.

Our first step is to compute the posterior distribution on $\mathrm{sign}(\boldsymbol{z})$, conditioned on the observation $\boldsymbol{z}'$.

**Algorithm 8:** PRCWat.Decode

**Input:** Watermarking key k, image $\boldsymbol{x}$, and estimated error $\sigma$
**Output:** Decoded message $\boldsymbol{m} \in \{0,1\}^k$ or None
**1** $\boldsymbol{z} \leftarrow$ Recover($\boldsymbol{x}$);
**2 for** $i \in [n]$ **do**
**3** $\quad s_i = \mathrm{erf}\left(\frac{z_i}{\sqrt{2\sigma^2(1+\sigma^2)}}\right)$;
**4** result $\leftarrow$ PRC.Decode(k, $\boldsymbol{s}$);
**5** Output result;

**Fact 1.** *If $z \sim \mathcal{N}(0,1)$ and $z' \sim \mathcal{N}(z, \sigma^2)$ then*

$$\mathbb{E}[\mathrm{sign}(z) \mid z'] = \mathrm{erf}\left(\frac{z'}{\sqrt{2\sigma^2(1+\sigma^2)}}\right).$$

*Proof.* The joint distribution of $(z, z')$ is

$$\begin{pmatrix} z \\ z' \end{pmatrix} \sim \mathcal{N}\left(\begin{pmatrix} 0 \\ 0 \end{pmatrix}, \begin{pmatrix} 1 & 1 \\ 1 & 1+\sigma^2 \end{pmatrix}\right).$$

Using the formula for the conditional multivariate normal distribution,[13] the distribution of $z$ conditioned on $z'$ is

$$z \sim N\left(\frac{z'}{1+\sigma^2}, \frac{\sigma^2}{1+\sigma^2}\right).$$

Therefore

$$\Pr[z \geq 0 \mid z'] = \Phi\left(\frac{z'}{\sigma\sqrt{1+\sigma^2}}\right),$$

where $\Phi$ is the cumulative distribution function of the standard normal distribution, so

$$\mathbb{E}[\mathrm{sign}(z) \mid z'] = 2\Pr[z \geq 0 \mid z'] - 1$$
$$= 2\Phi\left(\frac{z'}{\sigma\sqrt{1+\sigma^2}}\right) - 1$$
$$= \mathrm{erf}\left(\frac{z'}{\sqrt{2\sigma^2(1+\sigma^2)}}\right),$$

where we have used the fact that $\Phi(x) = (1 + \mathrm{erf}(x/\sqrt{2}))/2$. □

Recall from Algorithm 2 that, in the PRC case, we generate the $i$th bit of the PRC codeword $\mathrm{sign}(z_i)$ by XORing the $i$th bit of a vector satisfying the parity checks with a random $e_i \sim \mathrm{Ber}(\eta)$ variable and the $i$th bit of the one-time pad $\mathsf{otp}_i$. Therefore we have

$$\mathbb{E}[(-1)^{\mathsf{otp}_i \oplus e_i} \cdot \mathrm{sign}(z_i) \mid z_i'] = (-1)^{\mathsf{otp}_i} \cdot (1 - 2\eta) \cdot \mathrm{erf}\left(\frac{z_i'}{\sqrt{2\sigma^2(1+\sigma^2)}}\right).$$

Let

$$s_i = \mathbb{E}[(-1)^{e_i} \cdot \mathrm{sign}(z_i) \mid z_i'] = (1 - 2\eta) \cdot \mathrm{erf}\left(\frac{z_i'}{\sqrt{2\sigma^2(1+\sigma^2)}}\right).$$

for each $i \in [n]$. Let $a_{\boldsymbol{w}} = \prod_{j \in \boldsymbol{w}}(-1)^{\mathsf{otp}_j}$ and $\hat{s}_{\boldsymbol{w}} = \prod_{j \in \boldsymbol{w}} s_j$ for each $\boldsymbol{w} \in \boldsymbol{P}$. Then $(1 + a_{\boldsymbol{w}}\hat{s}_{\boldsymbol{w}})/2$ is the probability that $(-1)^{\mathsf{otp} \oplus \boldsymbol{e}} \cdot \mathrm{sign}(\boldsymbol{z})$ satisfies $\boldsymbol{w}$.

Our detector simply checks whether

$$\log \prod_{\boldsymbol{w} \in \boldsymbol{P}} \left(\frac{1 + a_{\boldsymbol{w}}\hat{s}_{\boldsymbol{w}}}{2}\right) = \sum_{\boldsymbol{w} \in \boldsymbol{P}} \log\left(\frac{1 + a_{\boldsymbol{w}}\hat{s}_{\boldsymbol{w}}}{2}\right)$$

is greater than some threshold. We set the threshold by computing a bound on the false positive rate.

---

[13]See, for instance, (Holt & Nguyen, 2023, Theorem 3).

### E.2 BOUNDING THE FALSE POSITIVE RATE: PROOF OF THEOREM 2

To compute a bound on the false positive rate of the detector, we use a "bounded from above" version of Hoeffding's inequality due to Fan et al. (2015):

**Fact 2** (Corollary 2.7 from Fan et al. (2015))**.** *Let $X_1, \ldots, X_r \in \mathbb{R}$ be independent random variables such that $\mathbb{E}X_i = 0$, $X_i \leq b_i$, and $\mathbb{E}X_i^2 \geq b_i^2$. Then*

$$\Pr\left[\sum_{i=1}^r X_i \geq \tau\right] \leq \exp\left(-\frac{\tau^2}{2\sum_{i=1}^r \mathbb{E}X_i^2}\right).$$

We are now ready to prove Theorem 2. We state the theorem for the PRC detector and decoder; note that this immediately implies the same result for the PRC watermark detector and decoder.

**Theorem 2.** *Let $n, t \in \mathbb{N}$ and $F > 0$. For any string $\boldsymbol{s} \in [-1, 1]^n$,*

$$\Pr_{\mathsf{k}\sim\mathsf{PRC.KeyGen}(n,t,F)}[\mathsf{PRC.Detect}(\mathsf{k}, \boldsymbol{s}) = \mathsf{True}] \leq F$$

*and*

$$\Pr_{\mathsf{k}\sim\mathsf{PRC.KeyGen}(n,t,F)}[\mathsf{PRC.Decode}(\mathsf{k}, \boldsymbol{s}) \neq \mathsf{None}] \leq F.$$

*Proof.* Observe that the use of testbits immediately implies that PRC.Detect has a false positive rate of at most $F$, i.e.,

$$\Pr_{\mathsf{k}\sim\mathsf{PRC.KeyGen}(n,t,F)}[\mathsf{PRC.Decode}(\mathsf{k}, \boldsymbol{s}) \neq \mathsf{None}] \leq F.$$

We therefore turn to analyzing the false positive rate of PRC.Detect. We adopt the notation from Appendix E.1, with

$$a_{\boldsymbol{w}} = \prod_{j\in\boldsymbol{w}}(-1)^{\mathsf{otp}_j} \text{ and } \hat{s}_{\boldsymbol{w}} = \prod_{j\in\boldsymbol{w}} s_j$$

for each $\boldsymbol{w} \in \boldsymbol{P}$.

By construction of the parity check matrix, the parity checks $\boldsymbol{w} \in \boldsymbol{P}$ are linearly independent. Since otp is uniformly random, it follows that the values $a_{\boldsymbol{w}}$ are independent and uniformly random from $\{-1, 1\}$. Therefore by Fact 2 it suffices to show that

$$\Pr_{\boldsymbol{a}\sim\{-1,1\}^r}\left[\sum_{\boldsymbol{w}\in\boldsymbol{P}}\log\left(\frac{1+a_{\boldsymbol{w}}\hat{s}_{\boldsymbol{w}}}{2}\right) \geq \sqrt{C\log(1/F)} + \frac{1}{2}\sum_{\boldsymbol{w}\in\boldsymbol{P}}\log\left(\frac{1-\hat{s}_{\boldsymbol{w}}^2}{4}\right)\right] \leq F$$

where $r$ is the number of parity checks in $\boldsymbol{P}$ and $C = \frac{1}{2}\sum_{\boldsymbol{w}\in\boldsymbol{P}}\log^2\left(\frac{1+\hat{s}_{\boldsymbol{w}}}{1-\hat{s}_{\boldsymbol{w}}}\right)$.

Let

$$f_{\boldsymbol{w}}(a_{\boldsymbol{w}}) := \frac{1+a_{\boldsymbol{w}}\hat{s}_{\boldsymbol{w}}}{2}.$$

for each $\boldsymbol{w} \in \boldsymbol{P}$. Since $\boldsymbol{a}$ is random, each $f_{\boldsymbol{w}}(a_{\boldsymbol{w}})$ is uniformly random from $(1 \pm \hat{s}_{\boldsymbol{w}})/2$.

Let

$$X_{\boldsymbol{w}} = \log f_{\boldsymbol{w}}(a_{\boldsymbol{w}}) - \frac{\log f_{\boldsymbol{w}}(1) + \log f_{\boldsymbol{w}}(-1)}{2}$$

and

$$\begin{aligned}
b_{\boldsymbol{w}} &= \max_{y\in\{-1,1\}}\log f_{\boldsymbol{w}}(y) - \frac{\log f_{\boldsymbol{w}}(1) + \log f_{\boldsymbol{w}}(-1)}{2} \\
&= \left|\frac{\log f_{\boldsymbol{w}}(1) - \log f_{\boldsymbol{w}}(-1)}{2}\right|.
\end{aligned}$$

Then $X_{\boldsymbol{w}} \leq b_{\boldsymbol{w}}$ and

$$\mathbb{E} X_{\boldsymbol{w}}^2 = \mathbb{E}|X_{\boldsymbol{w}}|^2$$
$$= \left| \frac{\log f_{\boldsymbol{w}}(1) - \log f_{\boldsymbol{w}}(-1)}{2} \right|^2$$
$$= b_{\boldsymbol{w}}^2.$$

Applying Fact 2, we find that

$$\Pr\left[ \sum_{\boldsymbol{w} \in P} \log f_{\boldsymbol{w}}(a_{\boldsymbol{w}}) \geq \tau + \sum_{\boldsymbol{w} \in P} \frac{\log f_{\boldsymbol{w}}(1) + \log f_{\boldsymbol{w}}(-1)}{2} \right] \leq \exp\left(-\tau^2/C\right)$$

where $C = 2 \sum_{\boldsymbol{w} \in P} \mathbb{E} X_{\boldsymbol{w}}^2$. By the definition of $f_{\boldsymbol{w}}$,

$$\frac{\log f_{\boldsymbol{w}}(1) + \log f_{\boldsymbol{w}}(-1)}{2} = \frac{1}{2} \log\left( \frac{1 - \hat{s}_{\boldsymbol{w}}^2}{4} \right)$$

and

$$\mathbb{E} X_{\boldsymbol{w}}^2 = \frac{1}{4} \log^2\left( \frac{1 + \hat{s}_{\boldsymbol{w}}}{1 - \hat{s}_{\boldsymbol{w}}} \right).$$

Therefore

$$\Pr\left[ \sum_{\boldsymbol{w} \in P} \log f_{\boldsymbol{w}}(a_{\boldsymbol{w}}) \geq \tau + \frac{1}{2} \sum_{\boldsymbol{w} \in P} \log\left( \frac{1 - \hat{s}_{\boldsymbol{w}}^2}{4} \right) \right] \leq \exp\left(-\tau^2/C\right)$$

where $C = \frac{1}{2} \sum_{\boldsymbol{w} \in P} \log^2\left( \frac{1 + \hat{s}_{\boldsymbol{w}}}{1 - \hat{s}_{\boldsymbol{w}}} \right)$. The claim follows by setting $\tau = \sqrt{C \log(1/F)}$. $\qquad\square$

### E.3 PRACTICAL UNDETECTABILITY

We have not yet discussed the extent to which our scheme is undetectable for practical image sizes. As observed by Christ et al. (2024), the undetectability of any watermarking scheme can be broken with enough samples and computational resources: Undetectability just means that the resources required to detect the watermark *without* the key scale super-polynomially with the resources required to detect the watermark *with* the key. And under the same assumptions as in Christ & Gunn (2024), our scheme is asymptotically undetectable for the right scaling of parameters. We refer to our scheme as "undetectable" because of this, and because our experiments on quality and detectability demonstrate that it is undetectable enough for the main practical applications. However, for the specific, concrete choices of parameters used in our experiments, undetectability is not guaranteed against motivated adversaries.

For the PRC watermark, there exists a brute-force attack on undetectability that runs in time $O(n^{t-1})$, counting queries to the generative model as $O(1)$, where $n$ is the dimension of the image latents and $t$ is the sparsity of parity checks which can be set by the user (larger $t$ decreases the robustness). This attack works by simply iterating over $t$-sparse parity checks until one used by the watermark is found. We did not attempt to optimize the attack, so it is possible that faster attacks could be found.

In our experiments we have $n = 2^{14}$ dimensional image latents, and we set $t = 3$ for most of our experiments demonstrating robustness. To ensure cryptographic undetectability, a better choice would be $t = \log_2(n)/2 = 7$. The watermark detector still works with $t = 7$ for non-perturbed images, but we choose $t = 3$ for most experiments because of the improved robustness. Note that $O(n^2)$ is far greater than the $O(1)$ time required to detect prior watermarks without the key, but *a motivated adversary can still break the undetectability of our scheme*. We therefore stress that our scheme, in its current form, should not be used for undetectability-critical applications such as steganography.

The reason there exists a relatively fast brute-force distinguishing attack against our scheme is that there exist quasi-polynomial time attacks against the PRC of Christ & Gunn (2024). The alternative constructions of PRCs due to Golowich & Moitra (2024) and Ghentiyala & Guruswami (2024) also suffer from quasi-polynomial time attacks. It is an interesting open question to construct PRCs that do not have quasi-polynomial time attacks; using our transformation, any such PRC would generically yield a watermarking scheme with improved undetectability. We hope that generative image model watermarks with improved undetectability can be built in the future.

Images without Watermark

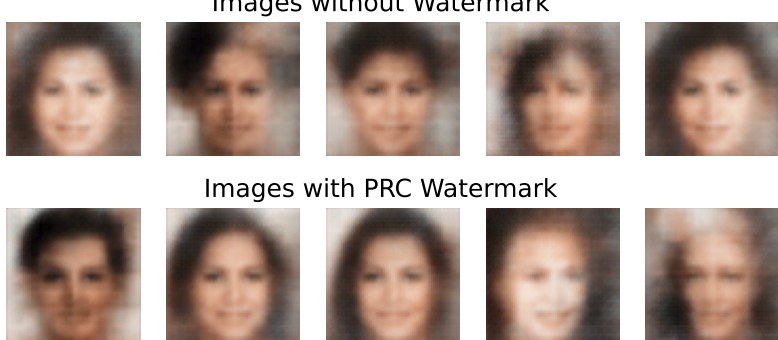

Images with PRC Watermark

Figure 12: Comparison of unwatermark and PRC watermark images on VAE.

## F    DEMO: PRC WATERMARK FOR VAES

The PRC watermark can be applied to VAEs (Kingma & Welling, 2013) as well. Using the same gradient descent technique as Hong et al. (2023), we optimize the latent to obtain the decoder inversion result for watermark detection. We test the PRC watermark on a VAE with a 256-dimensional latent space, trained on the CelebA dataset (Liu et al., 2018). By setting $t = 2$ and FPR as 0.05, we achieve over 90% TPR when embedding a zero-bit PRC watermark in the images. We show example generated images in Figure 12. We did not investigate the robustness or quality of the PRC watermark for VAEs in-depth, so this section is only to demonstrate the generality of our technique.

