# OpenReview forum: "An Undetectable Watermark for Generative Image Models"
_NeurIPS.cc/2024/Workshop/SafeGenAi — SafeGenAi Poster_

### Official Review · Reviewer_zHwG · 2024-10-11
**Thorough Experiment and Analysis**

**Rating:** 7
**Confidence:** 3

**Review:**

This study introduces a novel undetectable watermarking scheme for generative image models, effectively preventing detection by adversaries through adaptive queries. The authors use a pseudorandom error-correcting code to ensure both the undetectability and robustness of the watermark. Their experimental validation, conducted using Stable Diffusion, confirms that the watermark does not compromise image quality and is resistant to existing watermark removal techniques.

---

### Official Review · Reviewer_zVrz · 2024-10-12
**This paper introduces PRC Watermark, an undetectable watermarking scheme for generative image models. The method embeds pseudo-random patterns into the initial Gaussian noise distribution using PRC, ensuring both undetectability and quality preservation. The experiments demonstrate that (1) image quality is maintained, (2) watermarks remain undetectable, and (3) the watermark detector is robust against various attacks.**

**Rating:** 8
**Confidence:** 3

**Review:**

**Quality**
The overall quality of the work is high, and the theoretical background is well-stated. The experiments strongly support the claims made in the paper. One concern is that this paper uses specific Generate and Recover algorithms. I wonder what the results would be if the accuracy of the Recover algorithm is lower.

**Clarity**
The paper is well-written and easy to follow. Although most of the actual PRC algorithms are explained in the reference paper and the Appendix, the essence of the method is clearly shown in the main text. The experiment results are well-displayed and easy to understand.

**Originality**
Incorporating PRC into watermarking is a novel approach.

**Significance**
This method significantly outperforms previous approaches in terms of image quality, undetectability, and robustness. Especially, the undetectability shown in Figure 1 is impressive.

**Pros and Cons**
Pros: Good motivation, clear explanation, significant performance gain.
Cons: Applicability to different architectures